# Thrombospondin-1 CD47 Signalling: From Mechanisms to Medicine

**DOI:** 10.3390/ijms22084062

**Published:** 2021-04-14

**Authors:** Atharva Kale, Natasha M. Rogers, Kedar Ghimire

**Affiliations:** 1Centre for Transplant and Renal Research, Westmead Institute for Medical Research, Westmead, NSW 2145, Australia; akal8294@uni.sydney.edu.au; 2Westmead Clinical School, Faculty of Medicine and Health, The University of Sydney, Sydney, NSW 2145, Australia; 3Renal and Transplant Unit, Westmead Hospital, Westmead, NSW 2145, Australia; 4Starzl Transplant Institute, University of Pittsburgh, Pittsburgh, PA 15213, USA

**Keywords:** thrombospondin-1 (TSP1; THBS1), CD47 (IAP), matricellular, fibrosis, kidney injury, glucose homeostasis, ageing, reactive oxygen species

## Abstract

Recent advances provide evidence that the cellular signalling pathway comprising the ligand-receptor duo of thrombospondin-1 (TSP1) and CD47 is involved in mediating a range of diseases affecting renal, vascular, and metabolic function, as well as cancer. In several instances, research has barely progressed past pre-clinical animal models of disease and early phase 1 clinical trials, while for cancers, anti-CD47 therapy has emerged from phase 2 clinical trials in humans as a crucial adjuvant therapeutic agent. This has important implications for interventions that seek to capitalize on targeting this pathway in diseases where TSP1 and/or CD47 play a role. Despite substantial progress made in our understanding of this pathway in malignant and cardiovascular disease, knowledge and translational gaps remain regarding the role of this pathway in kidney and metabolic diseases, limiting identification of putative drug targets and development of effective treatments. This review considers recent advances reported in the field of TSP1-CD47 signalling, focusing on several aspects including enzymatic production, receptor function, interacting partners, localization of signalling, matrix-cellular and cell-to-cell cross talk. The potential impact that these newly described mechanisms have on health, with a particular focus on renal and metabolic disease, is also discussed.

## 1. Matricellular Proteins

Matricellular proteins consist of a group of diverse proteins which are dynamically expressed and secreted into the extracellular matrix (ECM). The term ‘matricellular’ was coined to explain the diversity of function attributed to these proteins. These proteins do not primarily provide structural integrity to the extracellular microenvironment, but rather perform a regulatory capacity within the ECM by interacting with cell-surface receptors, enzymes, and growth factors. These proteins also govern cell phenotype via interactions with various receptors and mediate cell-to-cell and cell-to-environment communication [1,2,3].

## 2. Thrombospondin-1 (TSP1)

The thrombospondins are a family of five glycoproteins with functional remodelling properties similar to other matricellular proteins, as they are all capable of mediating ECM organisation, and interact with growth factors and cytokines for cell-to-cell communication. They display comparable structural features that consist of well-defined binding domains that link to proteoglycans and integrins [4,5]. Thrombospondins have diverse physiological and pathophysiological tissue-specific actions that include effects on vascular responsiveness [6,7] and angiogenesis [8,9,10], platelet activation [11,12], inflammation [13,14], and cell death [15] that directly impact on wound healing and tumorigenesis.

Thrombospondin-1 (TSP1) is a multi-modular glycoprotein within the thrombospondin family, and was first described by Baenziger et al. nearly five decades ago [16]. TSP1 consists of an N-terminal (NH_2_) domain, three properdin type-I repeats, three epidermal growth factor-like type-II repeats, a calcium-binding type-III motif, and a C-terminal (CO-OH) domain that facilitates cell attachment [17,18]. The homotrimeric form is required for retention of TSP1 in the ECM [18] and in vitro effects [19]. A substantial, pre-formed residual pool of TSP1 is sequestered in α-granules of platelets, and this is released into the surrounding microenvironment following the activation of platelets by thrombin. Physiological levels of TSP1 in the blood plasma are ~100–150 pM [20,21,22,23]. Numerous cell types, including innate immune cells (dendritic cells [13] and macrophages [24]), and parenchymal cells (endothelial [7], vascular smooth muscle [25] and epithelial cells [26]) induce expression of TSP1 following injury or stress (Figure 1). Upregulation of TSP1 has been demonstrated by hypoxia [26] via hypoxia-inducible factor (HIF)-2α [27], glucose [28] (albeit followed by posttranscriptional downregulation), and LPS [29]. There are multiple additional transcription factors described that promote or inhibit TSP1 gene expression (expertly reviewed in [30]).

The multiple binding domains of TSP1 allow it to interact with an abundance of other proteins including receptors, integrins, proteases, ECM, and growth factors (Figure 1). TSP1 binds to heparin, LRP1 (low-density lipoprotein (LDL) receptor-related protein)/calreticulin, tumour necrosis factor-stimulated gene-6 (TSG6), β1 integrins, aggrecan, decorin and sulphated glycolipids through its N-terminal domain [31]. The type I repeats of TSP1 have been shown to bind to collagen V and VII, fibrinogen, jagged1, laminin, matrix metalloproteinases-2 and 9, TGF-β, CD36, and von Willebrand factor [31]. The type II repeats of TSP1 bind to voltage-dependent calcium channel subunit and α2δ-1 and β1 integrins, whereas the type III repeats bind to β3 integrins, cathepsin G, fibroblast growth factor 2, elastase and ionised calcium (Ca^2+^). TSP1 binds to receptor CD47 at the C-terminal domain [12,31,32].

This diverse interactome provides TSP1 with the capacity to execute multiple, context-specific roles ranging from modification of cellular function to remodelling the extracellular matrix. Many of these effects have also been demonstrated to be bi-phasic and dose-dependent, which can account for apparent discrepant findings in the literature. Recently, TSP1-CD47 signalling in RBCs was found to modulate calcium influx via voltage-gated ion channels and further evoke cellular deformity and death [33]. The scavenger receptor, CD36, mediates complex formation with TSP1 and non-receptor kinases such as Lyn and Fyn to orchestrate inflammation [34]. Syndecan-1, a proteoglycan, facilitates fascin spike formation in cells following binding of TSP1 and this enhances contact with matrix structures [35]. CD148 is a receptor-type protein tyrosine phosphatase involved in the downregulation of growth factors in cells and serves as a receptor for TSP1 to suppress cellular growth and proliferation [36]. The role of TSP1 also extends in neuronal biology: ligation of Apolipoprotein E receptor 2 and very low density lipoprotein receptor (VLDLR) mediates post-natal neuronal migration [37], and TSP1-neuroligin-1 interaction accelerates synaptogenesis in hippocampal neurons [38].

CD47, also known as integrin-associated protein (IAP), is a 50 kDa cell surface receptor. As the high-affinity receptor for TSP1 and binding at picomolar concentrations, it likely provides the dominant signalling moiety in vivo. It is ubiquitously expressed throughout the body, including red blood cells. It remains an unusual member of immunoglobulin (IgG) family of proteins in that it has a single extracellular immunoglobulin-like domain followed by five transmembrane helices [39]. CD47 binds both large glycoproteins such as TSP1 (480 kDa) and smaller cell-based ligands, including SIRPα (55 kDa), independent of its association with integrins [39]. In addition, CD47 interacts in a cis-cis formation with the surface receptor Fas/CD95, and modulates Fas-mediated apoptosis [15]. The C-terminal domain of TSP1 ligates the extracellular N-terminal domain of CD47 to initiate downstream signalling [40]. Extensive literature accrued over the last decade of research has demonstrated critical roles for TSP1-CD47 signalling in cardiovascular biology [41], cancer [42], and inflammation [32].

Many of the effects demonstrated by TSP1-mediated ligation of CD47 are also replicated by binding to CD36 (inhibition of endothelial cell migration [43], decreased VEGF phosphorylation [44]). However, there are marked differences in receptor affinity: CD47-based effects occur at physiological (picomolar) level whereas CD36-mediated interactions require 100-fold greater concentrations of TSP1 [45]. Under pathological and inflammatory conditions, TSP1 levels in tissue or plasma are elevated to concentrations that assume activation of receptors in addition to CD47.

## 3. Role of TSP1 Signalling in Inflammation

Inflammation is a routine, biological, and protective response of tissues to infection and cellular damage that involves recruitment of innate immune cells (e.g., macrophages and neutrophils) to the injured site, and subsequently governs modulation of cell phenotype to regulate recovery. Ligation of TSP1 to cell surface receptors CD36 and CD47 is associated with the regulation of inflammatory responses [46], although the literature demonstrates dichotomous findings dependent upon the disease models used. TSP1 limits leukocyte phagocytic capacity for fungi (particularly *Candida*), leading to increased fungal burden and disseminated infection [47]. TSP1-null mice demonstrate paradoxical responses to gram-negative bacteria, displaying an exaggerated inflammatory response that increases susceptibility to *Pseudomonas* [48], but more effective bacterial containment following pneumonia infection with *Klebsiella* [49]. Even in conditions that lack a bacterial infection as the primary inciting event, TSP1-null mice are more susceptible to severe acute colitis [50] including megacolon and peritonitis, as well as bleomycin-induced lung injury [51]. In both studies, replacing TSP1 or initiating receptor activation mitigated inflammatory responses.

Studies in mice have shown that TSP1 deficiency decreases the inflammatory phenotype of macrophages, which also crucially mediate parts of the innate immune system. TSP1 interaction with CD36 on macrophages regulates their activation and function, resulting in the stimulation of toll-like receptor 4 (TLR4) [52]. Increased transcriptional expression of TLR4 also augments the activity of nuclear factor-kappaB (NF-κB), a master regulator of pro-inflammatory cytokine production. Enhanced expression of tumour-necrosis factor (TNF)-α is observed in macrophages derived from bone-marrow of mice in a TSP1 dose-dependent manner [52]. These data support the notion that TSP1 also influences cytokine activity.

Studies have demonstrated elevated expression of TSP1 in inflamed adipose tissue of obese and insulin-resistant subjects [53,54]. TSP1 deficiency reduces obesity-associated inflammation and improves insulin sensitivity in a diet-induced (obese) wild-type C57BL/6J mouse model [53]. In pre-clinical studies, TSP1 deficiency did not affect the development of obesity in high-fat-fed mice, however, pro-inflammatory F4/80^+^CD11c^+^ macrophage recruitment to and accumulation in adipose tissue was significantly reduced. Analysis of adipose tissue and bone-marrow-derived macrophages demonstrated decreased mRNA transcript expression of pro-inflammatory cytokines (including TNF-α and IL-6) in obese TSP1-deficient mice. TSP1 via CD36 signalling has been shown to promote weight gain by enhancing adipocyte fatty acid uptake and stimulating adipocyte proliferation. TSP1-null mice fed a high-fat diet demonstrate lower insulin levels, suggesting amelioration of insulin resistance [54]. Furthermore, a lack of TSP1 has been shown to improve both glucose tolerance and insulin sensitivity [53], and reduce weight gain in high-fat diet-induced obese mice [53]. These findings strongly suggest that TSP1 upregulation in adipose tissue contributes to weight gain, inflammatory macrophage activation, and associated metabolic dysfunction [54].

## 4. TSP1-CD47 Signalling in Cardiovascular Diseases

Cardiovascular diseases encompass a broad range of pathologies affecting small vessels, such as coronary artery disease, and large vessels which contribute to hypertension. The biogas nitric oxide (NO) is a major regulator of vascular tone [55]. It is primarily synthesised in the endothelium, although it is produced under homeostatic conditions by several other cell types including vascular smooth muscle cells (VSMC) [56]. NO released by endothelium acts on VSMC and promotes vasodilation by activating soluble guanylyl cyclase (sGC) and consequent conversion of GTP to cGMP [57]. The resulting vasodilation increases blood flow and reduces blood pressure. TSP1 inhibits NO-stimulated vascular cell outgrowth, endothelial responses (proliferation, adhesion, chemotaxis) through engagement of receptors CD36 and CD47, however only CD47 engagement has been shown to be necessary for these activities of TSP1 at physiological (nanomolar) concentrations [45,58]. TSP1 binding to CD47, antagonises NO signalling by inhibiting sGC and preventing cGMP synthesis, allowing TSP1 to acutely constrict blood vessels [6,7]. Conversely, hypotension manifests in both TSP1- and CD47-null mice [59]. These mice displayed greater hypotensive response to NO-donor agents administered systemically compared to the littermate controls, regardless of the kinetics of NO-release, as well as blunted responses to traditional hypertensive agents, and elimination of central mechanisms that control blood pressure increase mortality [59].

In coronary artery disease such as atherosclerosis, there is loss of NO synthesis and an accumulation of apoptotic cellular debris and diseased vascular cells in the artery wall, which places advanced plaque lesions at risk of rupture [60]. TSP1 deficiency accelerates inflammatory changes required for plaque maturation and instability in susceptible (ApoE-deficient) mice [61]. However, this role is now disputed with an apparent protective phenotype in the context of hyperleptinemia [62]. CD47 has been found to be upregulated in human atherosclerotic plaques compared to non-atherosclerotic vascular tissue and administration of anti-CD47 antibodies ameliorates atherosclerosis by reversing defects in efferocytosis, normalizing the clearance of diseased vascular tissue [63]. Interestingly, the anti-atherosclerotic effects mediated by CD47 antibodies were independent of TSP1 signalling.

TSP1-CD47 signalling is upregulated in diseased human lung parenchyma and distal pulmonary arteries, and has been implicated in the development of pulmonary hypertension in mouse models of disease [7,64,65]. Intact TSP1-CD47 signalling contributes to pulmonary arterial vasculopathy and dysfunction in much the same manner that the systemic vasculature is affected: hampering vasodilation and promoting vasoconstriction [3]. The activation of CD47, via TSP1, stimulates the interaction of endothelin-1 (ET-1) with endothelin receptor A, mediating proliferation and hypertrophy of pulmonary arterial smooth muscle cells as well as vasoconstriction, leading to pulmonary hypertension [7]. Interestingly, intact TSP1-CD47 signalling in endothelial cells supresses the transcription factor c-Myc, a ubiquitous regulator of gene expression required for cell growth. Previous work identified had detected that this proto-oncogene binds to the ET-1 E-box motif, with biphasic regulation of gene expression in fibroblasts [66]. Our findings [7] extend the reach of this regulatory gene pathway, demonstrating its role in driving vascular pathology. TSP1 also uncouples endothelial NO synthase to promote generation of reactive oxygen species [64] and augments expression of hypoxia inducible factor-2α which contributes to pathophysiological changes in the pulmonary vasculature [27].

## 5. TSP1-CD47 Signalling in Angiogenesis and Wound Healing

NO-cGMP signalling present in endothelial cells promotes chemotaxis and proliferation, both of which are crucial for angiogenesis [67]. TSP1, via CD47, constitutively and ubiquitously inhibits NO signalling, achieving this in part by suppression of VEGFR2—the main receptor for vascular endothelial growth factor (VEGF) [68]. When CD47 expression was absent in human and murine endothelial cells, VEGFR2 phosphorylation was no longer inhibited and angiogenic responses improved [10]. A recent study revealed that the anti-angiogenic effects of CD47 escalate with age in humans and mice, defining this pathway as a potential therapeutic target in age-associated vascular impairments [8].

Wound healing is a sophisticated series of processes that include coagulation, inflammation, cellular migration, and proliferation, including angiogenesis. The clotting system plays a major initial role in wound healing [69] and the role of TSP1 in the regulation of thrombosis has been studied extensively. Under pathological, high-shear (arterial) conditions, secreted TSP1 promotes the aggregation and adhesion of platelets by binding to platelet glycoprotein Ibα (GPIbα). These bind von Willebrand factor (VWF) and play an essential role in thrombus formation at the site of vascular injury. Mice with TSP1 deficiency demonstrate a delayed rate of thrombosis compared to control animals. However, TSP1 does not have a demonstrable effect on initial platelet adhesion at the site of injury, regardless of the availability of VWF but it does impact upon thrombus growth [70].

Following vascular injury, collagen present in the sub-endothelial layers of the vasculature is exposed to turbulent blood flow [71]. TSP1 from activated platelets interact with the abundantly expressed CD47 and CD36 on platelets to promote their activation and adhesion, followed by thrombus formation on collagen. These interactions also maintain thrombus stabilization [12]. The underlying mechanism of platelet activation following TSP1-CD36 interaction involves the activation of Syk tyrosine kinase and a consequent Ca^2+^ signalling. As a result, αIIbβ3 integrin and ADP receptors are activated to promote platelet spreading. TSP1 binds to not only CD36, but also CD47 on platelet surface in a Ca^2+^-dependent manner to activate αIIbβ3 and promote platelet spreading [72]. The outside-in signalling of αIIbβ3 on platelets leads to a cascade of intracellular signalling events that mediate irreversible aggregation, stable adhesion, clot retraction, and cytoskeletal rearrangement, as well as subsequent thrombus growth [73]. TSP1-mediated platelet activation is also associated with a reduced cAMP signalling in platelets and haemostasis at the site of vascular injury. A genetic deletion of TSP1 in vitro did not have any impact on platelet activation, however, in vivo studies of TSP1-null mice demonstrated prolonged bleeding and defective thrombosis [74].

## 6. TSP1-CD47 Modulation of Reactive Oxygen Species (ROS) Signalling

ROS are essential biomolecules that play a crucial role in cellular physiology, but are also toxic moieties at high concentrations that promote pathophysiological processes [75]. Pathways that regulate ROS production are crucial to manage their potential toxicity. Activation of CD47, either by TSP1 or the CD47-specific TSP1-derived peptide 7N3, has been shown to increase the production of ROS in cells. TSP1, via CD47, stimulates ROS production in VSMCs and promotes vascular dysfunction by inducing oxidative stress [76]. In renal tubular epithelial cells, TSP1 stimulates ROS production by ligating SIRPα, a known co-ligand of CD47 [25]. SIRPα ligation induces phosphatidylinositol 3-kinase-dependent Rac1 recruitment to Nox1, increasing superoxide production [77]. TSP1-CD47 signalling has recently been shown to promote ROS production in human pulmonary artery endothelial cells, playing an active role in sickle cell-associated vasculopathy and the subsequent development of pulmonary hypertension [65]. In tumour immunotherapy, CD47 ligation by anti-CD47 antibodies or TSP1 induces G1-phase cell cycle arrest in Epstein-Barr virus (EBV)-transformed B cells through ROS generation, suggesting an appealing therapeutic intervention in EBV-associated tumours [78]. However, the role of TSP1-mediated ROS signalling in metabolic diseases is yet to be investigated.

## 7. TSP1-CD47 Signalling in Cell Death, Senescence, and Ageing

Ageing-associated illnesses such as cardiovascular disease, cancer, osteoarthritis, type 2 diabetes, and hypertension are substantial causes of morbidity and mortality in both the developed and developing world. Ageing is accompanied by persistent pro-inflammatory responses, altered metabolic function, and limited vascular responsiveness. Cellular mechanisms that promote these manifestations may be a suitable target for therapeutic intervention to mitigate the ageing process. A recent study showed that TSP1, via CD47, induced Nox1-derived ROS production which was responsible for p53-p21^cip^-Rb signalling and endothelial senescence in human pulmonary vessels [79]. Thus, TSP1-CD47 signalling in cells not only supresses angiogenesis but also potentiates vascular pathology by inhibiting the cell cycle and inducing a senescent cellular phenotype [80]. The ageing process alone increases TSP1 and CD47 levels in the vasculature of both humans and mice, leading to age-associated impairment of vital cellular processes, including blood flow, angiogenesis, cellular self-renewal, and glucose homeostasis [8]. These manifestations were significantly improved in CD47-deficient mice suggesting that pharmacologic inhibition of CD47 could be useful in treating ageing-associated disorders.

## 8. TSP1-CD47 Signalling in Cellular Self-Renewal and Stemness

The Yamanaka transcription factors (Oct3/4, Sox2, Klf4, c-Myc, also abbreviated as OSKM) are highly expressed in embryonic stem cells and pluripotent cells, and self-renewal can be induced in both mouse and human somatic cells in response to forced overexpression [81]. Remarkably, inhibition of CD47 signalling singlehandedly elevates the level of OSKM factors in several human and mouse cells [82,83]. CD47 depletion allows sustained proliferation of primary murine endothelial cells, increases asymmetric division, and enables these cells to spontaneously reprogram to form multipotent embryoid body-like clusters. The ageing process reduces the expression of these factors in human and mice vessels, but their expression can be maintained if CD47 is inhibited [8]. Enhanced OSKM expression following absent or reduced CD47 expression is replicated in several cell types, including renal tubular epithelial cells that facilitate repair from ischemia-reperfusion injury (IRI) [83]. Similarly, intestinal epithelial cell renewal was found to be restricted by the upregulation of CD47 in human and mice with colitis [84]. A deficiency of CD47 upregulated c-Myc in these cells and promoted cellular proliferation and migration.

## 9. TSP1-CD47 Signalling in Blood Disorders

TSP1-CD47 signalling has also been implicated in complications associated with sickle-cell disease (SCD). SCD results from a point genetic mutation that substitutes glutamic acid for valine at position 6 of the β-chain of haemoglobin, leading to polymerization of deoxygenated haemoglobin and distortion (‘sickling’) of red blood cells (RBCs) [85]. Such inflexibly-shaped cells cannot pass through the microcirculation efficiently, resulting in mechanical obstruction of vessels. Recurrent vaso-occlusion manifests as pain crises, and the combination of persistent haemolysis, anaemia and hypoxia contribute to the development of pulmonary hypertension, which is associated with increased mortality risk [86]. Transgenic SCD (Berk) mice were found to spontaneously develop pulmonary hypertension and right ventricular hypertrophy, and CD47-null mice transplanted with Berk marrow had decreased congestion of RBC due to limited TSP1 signalling [65]. TSP1 and CD47 expression was elevated in the pulmonary tissue of SCD mice and humans with pulmonary hypertension. SCD patients with vaso-occlusive episodes also demonstrated high plasma TSP1 levels [23].

## 10. TSP1-CD47 Signalling in Metabolic Disease

TSP1 is released in response to activation of cell stress-response pathways. Hyperglycaemia is both a driver and consequence of metabolic stress [41] and the TSP1 promoter region is known to contain a glucose response element [87]. Several studies have reported increased TSP1 transcription in endothelial, VSMCs and mesangial cells in response to elevated glucose [88,89].

Obesity predisposes humans to the development of diabetes mellitus and is a well-known factor in the development of metabolic syndrome, cardiovascular disease, and chronic kidney disease (CKD). The role and complexity of TSP1-based signalling in renal fibrosis has been expertly reviewed elsewhere [90]. TSP1 and CD47 have been implicated in the development of CKD [20], the former independently mediating fibrosis by activating the latent form of transforming growth factor-β (TGF-β) [91,92]. Both a global lack of CD47 expression or CD47 antibody blockade limit the production of TSP1 and development of renal interstitial fibrosis following an ischemic injury [20]. TSP1 and TGF-β expression and activity are simultaneously upregulated in the glomerular mesangial cells of the kidneys of high-fat diet-induced (obese) C57BL/6J mice and contribute to renal fibrosis by enhancing the production of extracellular matrix proteins [93]. Peptidic blockade of TSP1-TGF-β binding (by LSKL) mitigated interstitial fibrosis in rat models of unilateral ureteric obstruction [94] and mesangial proliferative glomerulonephritis [91]. TSP1-deficient mice are protected from diabetes-induced glomerular injury and podocyte loss [95]. The tubular atrophy and capillary rarefaction that characterises CKD may correlate with the anti-angiogenic function of TSP1. Short hairpin RNA limiting TSP1 transcription has been shown to increase VEGF and peritubular capillary density with concurrent decreases in matrix deposition in a mouse renal fibrosis model [96].

Renal fibrosis can develop independently of TGF-β. Free-fatty acids, which circulate at higher frequency in diabetes in the context of inadequate glycaemic control, stimulate the production of TSP1 in podocytes via the MAPK pathway. TSP1 via CD36 also induces podocyte apoptosis and dysfunction independent of TGF-β activity [97]. Similarly, glomerular hypertrophy, albuminuria, and pro-inflammatory associated with the development of diabetes and dysglycemia fail to develop in TSP1-null mice fed with high-fat diet [93]. Leptin is associated with obesity and mesangial cells obtained from wild-type mice display enhanced TSP1, fibronectin, collagen IV and TGF-β expression following leptin-treatment [93]. Leptin-induced effects disappear in TSP1-deficient mesangial cells.

The involvement of TSP1 has also been documented in the pathogenesis of diabetic cardiomyopathy [98]. Studies in obese db/db diabetic mice have demonstrated upregulation of TSP1 accompanied with an increased fibrosis in the cardiac interstitial tissue via the suppression of matrix metalloproteinases and increased TGF-β levels [99]. Type 1 diabetic Wistar Kyoto rats treated with a TSP1 antagonist—LSKL peptide—displayed improved cardiac function with no development of cardiac fibrosis by suppressing TGF-β activation [100].

TSP1 is also overexpressed in adipocytes from epididymal white adipose tissue and blood of high-fat diet-fed Wistar rats compared to the standard chow-diet (control) group. Adipocytes from these rats were also found to be insulin-resistant compared to the control, indicated by a lower insulin-stimulated glucose uptake. Under hyperglycaemic conditions, a post-translational regulation pattern was observed in cultured adipocytes of high-fat diet-fed rats where TSP1 expression decreased but TSP1 secretion increased [101]. The contribution of TGF-β to the development of obesity, metabolic syndrome and insulin resistance is well documented, and its activity is increased in adipocytes by TSP1 [101,102]. Recent studies demonstrated improved glucose tolerance and insulin sensitivity in CD47-null mice, which was maintained with age [8,103]. As ageing increases the expression of TSP1 and CD47, upregulated activity of this pathway may play a significant role in age-induced metabolic dysfunction [8]. Pancreatic endothelial cells express TSP1 and regulate β-cell function through activation of TGF-β [104]. Inhibition of TSP1 in islets intended for transplantation has also been suggested as a feasible strategy to improve islet graft revascularization and function [105]. CD47 is also involved in the regulation of hepatic inflammation and lipid metabolism and interestingly, its deficiency exacerbates steatohepatitis following exaggerated suppression of sirtuin 1 and peroxisome proliferator activated receptor alpha [106]. Comprehensive studies to examine molecular mechanisms of TSP1-CD47 modulated metabolic dysfunction remains to be studied.

## 11. Concluding Remarks 

Over the past 50 years, research on TSP1 signalling has advanced critical knowledge of its numerous roles in oncology and cardiovascular biology. However, sufficient focus has not been given at pre-clinical and clinical levels to characterize the nature and function of this pathway in metabolic diseases, which are proving to be a major global healthcare burden. While therapies that directly target TSP1 are yet to advance from pre-clinical studies into human trials, agents that disrupt known TSP1 receptors are an important avenue of investigation. The vital function of TSP1 in regulating cellular physiology and cytoprotection, particularly redox control, inflammation, and self-renewal has been well-documented (Figure 2). Given the capacity of cells to initiate TSP1 synthesis under injurious stimuli, targeting the ligand would prove challenging. TSP1 requires receptor binding to exert its pathogenic effects, and developing selective TSP1-related anti-receptor therapeutics are attractive strategies for future drug development. Indeed, anti-CD47 therapy is now recognised as a major therapeutic arm for treatment of malignancy, and this could be adapted for other diseases which demonstrate pathological TSP1 signalling through CD47.

## Figures and Tables

**Figure 1 ijms-22-04062-f001:**
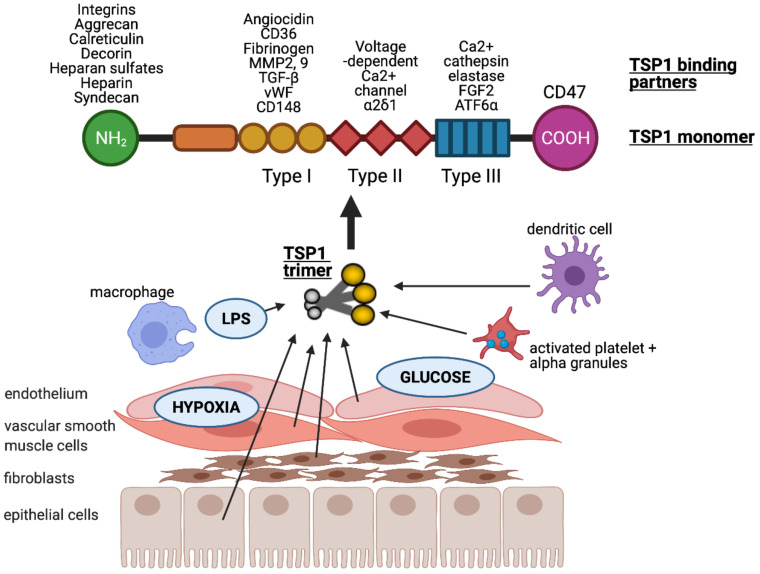
TSP1 is produced by many nucleated cells, including antigen presenting cells (macrophages and dendritic cells) and parenchymal cells (endothelial, vascular smooth muscle and epithelial cells, as well as fibroblasts). TSP1 is upregulated in response to injurious stimuli such as low oxygen tension (hypoxia), hyperglycaemia and lipopolysaccharide (LPS). It is typically secreted in trimeric form which is required for activity. Each monomer consists of multiple binding domains capable of interacting with a wide variety of factors.

**Figure 2 ijms-22-04062-f002:**
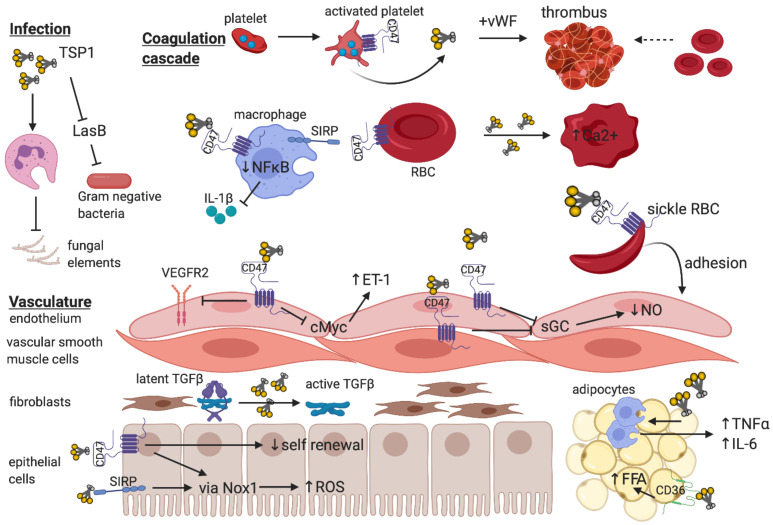
The breadth of TSP1 regulation of cellular function—The trimeric matricellular protein TSP1 is normally sequestered within platelet α-granules, but is released following platelet activation to promote thrombus formation at the site of vascular injury [70]. Within the circulation, TSP1 binds to CD47-expressing macrophages to limit NFκB-based transcription and release of pro-inflammatory cytokines [107] and inhibits virulence factors that facilitate gram negative bacterial infections [33]. Paradoxically, TSP1 impedes leukocyte phagocytosis of fungal elements thereby promoting susceptibility to infection [31]. Signal inhibitory regulatory protein (SIRP)-bearing patrolling macrophages also provide recognition of “self” via CD47 expressed on red blood cells (RBC), which inhibits phagocytosis [108]. Exogenous TSP1 disrupts RBC shape and increases intracellular Ca^2+^ [33]. Increased TSP1 released in the context of vaso-occlusive crises in sickle cell disease promote adhesion of CD47-expressing sickle RBC to endothelium [65]. Endothelial and vascular smooth muscle cells express CD47, and binding of TSP1 inhibits soluble guanylate cyclase (sGC) to reduce nitric oxide (NO) synthetic capacity [58] and limit vasorelaxation. Activated CD47 on endothelium inhibits VEGFR2 neo-angiogenic function [10] and downregulates cMyc expression to enhance endothelin-1 (ET-1) production [7] which is implicated in pulmonary hypertension. Fibroblasts synthesize matrix, and activation of latent TGF-β requires TSP1 independent of CD47 [91]. In parenchymal epithelial or vascular smooth muscle cells, TSP1 binding to either CD47 [76] or SIRP [25] induces ROS production via NADPH oxidase, and the former limits self-renewal capacity following injury [69]. TSP1 alone regulates inflammatory cell (macrophage) infiltrate into, and pro-inflammatory cytokine production by adipocytes, and binding to CD36 promotes free fatty acid uptake [39].

## Data Availability

Not applicable.

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
