# Peer review of "Thrombospondin-1 CD47 Signalling: From Mechanisms to Medicine"

_ijms, 2021, doi:10.3390/ijms22084062_

Round 1

Reviewer 1 Report

This review nicely summarizes the manifold roles of TSP-1 in a large number of physiologically very relevant processes. The authors carefully reviewed the current literature, compiled an impressive amount of information and provide a great overview on the TSP1-CD47 axis.

I only have a couple of comments that the authors might want to consider and that could futher improve the mansucript:

I can fully understand that the authors focus here on CD47. However, there are of course also other receptors that interact with TSP-1 (like e.g. CD36 and (several) integrins). Maybe a short paragraph briefly mentioning all potential receptors could help.

In some paragraphs the role of TSP1-CD36 signalling is described and sometimes it even seems that, for distinct processes, this interaction is more important than the CD47 interaction. For example, in paragraph 3 on inflammation, mostly CD36 is discussed. This might confuse the reader but a clear statement (what different (types of) receptors exist) at the beginning of the review might help. Maybe even think about an extra paragraph.

I also had the feeling that it could be interesting to clearly mention the different cell types that do express TSP1. From Figure 1, it looks as if platelets are the only or major source which might not be true.

According to the title, the reader might expect a short summarizing statement or future perspectives on therapeutic options/targets. What exactly would be targeted how: the ligand, the receptor, a distinct signaling pathway? Peptides, antibodies, small compounds?

line 71: a reference for this statement (that rimerization is needed) would be good

Keywords: I am uncertain if TSP1 AND TSP-1 is needed

Author Response

The authors wish to thank the reviewer for their comments.

I can fully understand that the authors focus here on CD47. However, there are of course also other receptors that interact with TSP-1 (like e.g. CD36 and (several) integrins). Maybe a short paragraph briefly mentioning all potential receptors could help.

Response: Thank you for this comment. We do recognise our intrinsic bias towards TSP1-CD47 interactions. The original manuscript had outlined the majority of  TSP1-binding partners (page 5). We have added a diagram (now Figure 1) and additional text to optimise clarity of this description.

In some paragraphs the role of TSP1-CD36 signalling is described and sometimes it even seems that, for distinct processes, this interaction is more important than the CD47 interaction. For example, in paragraph 3 on inflammation, mostly CD36 is discussed. This might confuse the reader but a clear statement (what different (types of) receptors exist) at the beginning of the review might help. Maybe even think about an extra paragraph.

Response: This is a good point. Many effects of TSP1-CD47 effects are also described in the TSP1-CD36 literature. We add to the text on pages 5-6. However, CD47 is clearly the high affinity receptor, as outlined in that paragraph.

I also had the feeling that it could be interesting to clearly mention the different cell types that do express TSP1. From Figure 1, it looks as if platelets are the only or major source which might not be true.

Response: Thank you for identifying that the Figure and text lack clarification. The original manuscript outlined the panoply of cells that also produce TSP1 (pages 4-5) however we have clarified this in the text and added a figure.

According to the title, the reader might expect a short summarizing statement or future perspectives on therapeutic options/targets. What exactly would be targeted how: the ligand, the receptor, a distinct signaling pathway? Peptides, antibodies, small compounds?

Response: We have adjusted the Conclusion to reflect future perspectives.

line 71: a reference for this statement (that dimerization is needed) would be good

Response: We have expanded on this (briefly) and added references.

Keywords: I am uncertain if TSP1 AND TSP-1 is needed

Response: we have kept one key word (TSP1, as per the text)

Reviewer 2 Report

The authors did a great work with their review article of TSP1 CD47 signaling. Some edits that could improve the manuscript are suggested below.

Minor:

Revise lines 75-47 on page 2. Having them on top of Figure 1 make them go missing and affect readability. I recommend to either move them below Fig1 or move Figure 1 one paragraph down so it is visually clear that there are text above it.

Major:

Figure 1 is convoluted and it is really hard to follow the info in the Legend with the flow of the picture. Authors may want to redesign. Also legend starts with platelets alpha-granules (top of figure), but the only label of the pictorial trimmer is under “infection” on the far right of the figure.

I suggest replacing or repurposing the “Concluding Remarks” section to highlight the views of the authors for future research directions.

Author Response

Minor:

Revise lines 75-47 on page 2. Having them on top of Figure 1 make them go missing and affect readability. I recommend to either move them below Fig1 or move Figure 1 one paragraph down so it is visually clear that there are text above it.

Response: We have changed the location of the Figure (and added another) to assist with the flow of the text.

Major:

Figure 1 is convoluted and it is really hard to follow the info in the Legend with the flow of the picture. Authors may want to redesign. Also legend starts with platelets alpha-granules (top of figure), but the only label of the pictorial trimmer is under “infection” on the far right of the figure.

Response: We have provided another figure that outlines production of TSP1 by different cell types and binding partners. We feel that Figure 2 (the original figure 1) is very comprehensive but the addition of another figure reduces the confusion.

I suggest replacing or repurposing the “Concluding Remarks” section to highlight the views of the authors for future research directions.

Response: We have altered the Conclusion to highlight future directions in the field. However, given that we are limited by the lack of progress in the field of TSP1-based therapeutics we are reluctant to dwell on simple conjecture.